# Mechanisms and impact of public reporting on physicians and hospitals' performance: A systematic review (2000–2020)

Khic-Houy Prang[1]*, Roxanne Maritz[1,2,3], Hana Sabanovic[1], David Dunt[1], Margaret Kelaher[1]

1 Centre for Health Policy, Melbourne School of Population and Global Health, The University of Melbourne, Carlton, Australia, 2 Rehabilitation Services and Care Unit, Swiss Paraplegic Research, Nottwil, Switzerland, 3 Department of Health Sciences and Health Policy, University of Lucerne, Lucerne, Switzerland

* khic-houy.prang@unimelb.edu.au

**Data Availability Statement:** All relevant data are within the manuscript and its Supporting Information files.

## Abstract

### Background

Public performance reporting (PPR) of physician and hospital data aims to improve health outcomes by promoting quality improvement and informing consumer choice. However, previous studies have demonstrated inconsistent effects of PPR, potentially due to the various PPR characteristics examined. The aim of this study was to undertake a systematic review of the impact and mechanisms (selection and change), by which PPR exerts its influence.

### Methods

Studies published between 2000 and 2020 were retrieved from five databases and eight reviews. Data extraction, quality assessment and synthesis were conducted. Studies were categorised into: user and provider responses to PPR and impact of PPR on quality of care.

### Results

Forty-five studies were identified: 24 on user and provider responses to PPR, 14 on impact of PPR on quality of care, and seven on both. Most of the studies reported positive effects of PPR on the selection of providers by patients, purchasers and providers, quality improvement activities in primary care clinics and hospitals, clinical outcomes and patient experiences.

### Conclusions

The findings provide moderate level of evidence to support the role of PPR in stimulating quality improvement activities, informing consumer choice and improving clinical outcomes. There was some evidence to demonstrate a relationship between PPR and patient experience. The effects of PPR varied across clinical areas which may be related to the type of indicators, level of data reported and the mode of dissemination. It is important to ensure that the design and implementation of PPR considered the perspectives of different users and the health system in which PPR operates in. There is a need to account for factors such

**Funding:** This work was supported by Medibank Better Health Foundation. The funders had no role in study design, data collection and analysis, decision to publish, or preparation of the manuscript.

as the structural characteristics and culture of the hospitals that could influence the uptake of PPR.

## Introduction

It is becoming increasingly common for healthcare systems internationally to measure, monitor and publicly release information about healthcare providers (i.e. hospitals and physicians) for greater transparency, to increase accountability, to inform consumers' choice, and to drive quality improvement in clinical practice [1–3]. In theory, public performance reporting (PPR) is hypothesised to improve quality of care via three pathways: selection, change and reputation.

- In the *selection* pathway, consumers compare PPR data and choose high-quality providers over low-quality providers, thereby motivating the latter to improve their performance.

- In the *change* pathway, organisations identify underperforming areas, leading to performance improvement. These pathways are interconnected by providers' motivation to maintain or increase market share [4].

- In the *reputation* pathway, PPR can negatively affect the public image of a provider or an organisation. Reputational concerns will therefore motivate providers or organisations to protect or improve their public image by engaging in quality improvement activities [5].

Given these different pathways, it is therefore not surprising that the measurement of PPR is complex. The quality indicators used (e.g. healthcare structure, processes, and patient outcomes), the mode of data publications (e.g. report cards) and the level of reporting (e.g. physician, unit or hospital level) vary widely across different healthcare systems and countries [6,7]. For example, in the United States (US) and the United Kingdom (UK), quality indicators such as mortality, infection rates, waiting times and patient experience are reported in the form of star ratings, report cards and patient narratives at the hospital and individual physician levels [8,9]. In Australia, performance of all public hospitals is publicly reported on the MyHospitals website [10]. Quality indicators reported include infections rates, emergency department waiting times, cancer surgery waiting times and financial performance of public hospitals. Reporting to MyHospitals is mandatory for Australian public hospitals but voluntary for private hospitals. Australia does not currently report at the individual physician level [11,12].

Research on the impact of PPR though is growing, as characterised by the large number of reviews published [7,13–22]. Previous reviews suggest that PPR has limited impact on consumers' healthcare decision-making and patients' health outcomes [16,22]. In contrast, there is evidence that PPR exerts the greatest effect among healthcare providers by stimulating quality improvement activities [13,15,23].

Yet, the effects of PPR on healthcare processes, consumers' healthcare choice and patients' outcomes still remain uncertain or inconsistent. For example, PPR affects consumers' selection of health plans but not selection of individual physicians or hospitals [13,15,20]. This may be because consumers do not always perceive differences in quality of healthcare providers, and they do not trust or understand PPR data [23,24]. Furthermore, it is often not clear how consumers' healthcare choices are constrained by systems-level (e.g. lack of choice due to geographical distance) and socio-cultural barriers (e.g. poor consumer health literacy). This uncertainty reflects the complexity surrounding PPR including the different healthcare choices consumers are asked to make and how this can ultimately influence various health outcomes.

Further, considering healthcare providers behaviours and quality improvement, there is some discrepancy on this position [16,22]. The discrepancy among the reviews likely reflects the complexity with various characteristics of PPR examined. For example, some reviews focused on the mechanisms by which PPR exerts influence [7,15] without differentiating between the heathcare choices consumers are asked to make, while others focused on impact [18,19] with the inclusion of a variety of patients outcomes across a range of healthcare settings or conditions. Furthermore, issues in the design and implementation of PPR (e.g. level of reporting, indicators and dissemination), type of audiences (e.g. consumers, providers, and purchasers) and primary purposes (e.g. selection of physician or hospital and change in clinical processes), are likely to lead to different effects (Table 1).

As a point of departure from previous reviews, the goal of this systematic review was to address these discrepancies. It does so by differentiating the effects of PPR by users and providers across various healthcare settings and conditions to provide greater conceptual clarity surrounding the impacts and utility of PPR. Therefore, the aim of this systematic review was to provide an updated evidence summary of the impact of PPR on physicians and hospitals' performance, focusing on the mechanisms (selection and change pathways) by which PPR exerts its influence.

## Methods

The study was conducted as part of a wider review of the impacts of PPR on outcomes among healthcare purchasers (public and private), providers (organisations and individual physicians) and consumers. The results of the other parts of the wider review are reported elsewhere [20,21]. The review was performed in accordance with the Preferred Reporting Items for Systematic Reviews and Meta-Analyses (S1 Checklist) guidelines [25].

### Search strategy

Five databases were searched from their dates of inception to 16th April 2015: Medline; Embase; PsycINFO; the Cumulative Index to Nursing and Allied Health Literature (CINAHL); and Evidence-Based Medicine Reviews (EBMR). The search strategy was based on Ketelaar et al. [16] (limited to experimental study designs) and extended to include observational study designs if they conformed to the Meta-analysis of Observational Studies in Epidemiology guidelines (MOOSE) [26]. Search terms were amended with the assistance of a librarian (see S1 Appendix for Medline search strategy). Results of searches were downloaded into Endnote X9.

**Table 1. Classification of public performance reporting by mechanisms and audiences.**

| Quality improvement (performance) | Organisation or practitioner subject to PPR | Performance measures subject to PPR |
|---|---|---|
| | Hospitals (units in hospitals)<br>Medical specialists<br>Health plans (e.g. HMOs)<br>Family physicians (general practitioners) | Clinical indicators<br>Structure indicators<br>Process indicators<br>Treatment indicators<br>Patient outcomes (e.g. mortality)<br>Patient experience |
| **Selection (services)** | **Consumer, organisation, or practitioner responding to PPR** | **Choice of services** |
| | Consumers<br>Hospitals (units in hospitals)<br>Medical specialists<br>Health plans (e.g. HMOs)<br>Family physicians (general practitioners) | Hospitals (units in hospitals)<br>Medical specialists<br>Health plans (e.g. HMOs)<br>Family physicians (general practitioners) |

HMO health maintenance oganisation; PPR public performance reporting.

A second search of the databases above was conducted on 14th November 2016 to include non-standard epidemiological descriptors (e.g. health economics literature) as previous search did not capture such studies: experimental studies; non-randomised studies; observational cohort; time trend; and comparative studies. Articles from previous systematic reviews on PPR were also screened [6,13,15–17,27,28]. A third search of the databases above was conducted on 3rd April 2020 to include additional studies published from 2016 to 2020.

## Inclusion and exclusion criteria

Articles were included if: 1) they examined the effect of PPR on outcomes among purchasers, providers or consumers; and 2) the study design was observational or experimental. Articles were excluded if: 1) performance reporting was not publicly disclosed; 2) they reported hypothetical choices; 2) the study design was qualitative; 3) it was published in languages other than English; 4) it was published prior to the year 2000 as the practice of PPR has change significantly since then due the widespread use of online PPR; 5) where pay-for-performance effects were not disaggregated from PPR; 6) they involved long-term care (e.g. nursing homes); and 7) studies perceived to be of low methodological quality following risk of bias assessment.

Two authors independently screened titles and abstracts for relevance and then assessed the eligibility of the full-text articles using a screening guide adapted from a previous meta-analysis [29] (see S2 Appendix). The methodological quality assessment was then conducted on the final selection of eligible full-text articles by two authors. Discrepancies between authors were discussed between them and if they remained unresolved, a third author made the final decision.

## Methodological quality assessment

The methodological quality of observational studies was assessed with the Newcastle-Ottawa Scale (NOS) [30] and RCT studies with the Cochrane Collaboration's tool for assessing risk of bias [31]. The NOS uses a star system based on three domains: the selection of the study groups; the comparability of the groups; and the ascertainment of either the exposure/outcome of interest. The Cochrane Collaboration's tool uses six domains to evaluate the methodological quality of RCT studies: selection bias; performance bias; detection bias; attrition bias; selective reporting; and other sources of bias. The methodological quality of each study was graded as low, moderate or high (see S3 Appendix). For cohort and quasi-experimental studies, a maximum of nine stars can be awarded: nine stars was graded as high methodological quality; six to eight stars as moderate methodological quality; and less than five stars as low methodological quality. For cross-sectional studies, a maximum of 10 stars can be awarded: nine to 10 stars was graded as high methodological quality; five to eight stars as moderate methodological quality and less than four stars as low methodological quality.

## Data extraction and synthesis

The following information was extracted from the articles: authors; year of publication; country; study design; study population; sample size; type of PPR data; outcome measures; statistical analysis; and findings including estimates. Studies considered to be of low methodological quality were excluded from the synthesis, however the characteristics and main findings of these studies are available in S4 Appendix. Given the high level of methodological heterogeneity and the heterogeneity of outcomes between the studies, no meta-analysis was performed. Instead, a systematic critical synthesis of the moderate and high methodological quality studies based on S1 Checklist guidelines was conducted. The strength of the evidence was determined using a rating system similar to that used in previous similar systematic reviews [7,19]. We

defined a positive effect in favour of PPR. We considered strong evidence if all studies showed significant positive effects, moderate evidence if more than half the studies showed significant positive effects, low evidence if a minority of studies showed significant positive effects, and inconclusive evidence if there were inconsistent findings across the studies (i.e. half of the studies showed significant positive effects and the other half significant negative effects) or insufficient findings (i.e. less than two studies).

## Results

### Inclusion of studies and quality assessment

In the first and second search, 8,627 articles were identified from five databases and eight previous reviews, resulting in 5,961 articles following removal of duplicates and those published prior to 2000 (Fig 1). In the third search, an additional 12,087 articles were identified from five databases, resulting in 9,603 articles following removal of duplicates. A total of 15,564 titles and abstracts were screened, with 15,447 articles excluded, leaving 117 articles for full-text screening. Following full-text screening, a total of 74 articles were included in the synthesis (59 and 15 articles from the previous searches and third search, respectively). Articles were categorised into three groups: 1) health plans; 2) coronary artery bypass graft (CABG) and percutaneous coronary intervention (PCI) and; 3) physicians and hospitals' performance. In this paper, results of physicians and hospitals' (n = 45) performance are presented. Nine studies were rated as high methodological quality and the rest as moderate methodological quality. The results are presented by mechanisms and impact of PPR:

- user and provider responses to PPR (selection of patients, physicians and hospitals including adverse selection, and organisational quality improvement) and

- impact of PPR on quality of care (improvement in clinical outcomes and patient experiences).

   Seven studies examined both the mechanisms and impact of PPR and are therefore included in both sections [32–38].

### Description of studies

Characteristics of the 45 studies are described in Table 2. Of these, nine studies examined the selection of patients, physicians and hospitals [39–47], 15 examined organisational quality improvement [48–62], and 14 examined the impact of PPR [63–76]. Seven studies investigated both user and provider behaviours to PPR and the impact of PPR [32–38]. All studies were published between 2002 and 2020. All studies were published in academic journals, except for three studies which were PhD dissertations [51,71,75]. Studies were predominantly conducted in the US (n = 26), followed by five from China, two from Canada, Japan, the Netherlands, and the UK, one from Australia, Germany, India, Italy, Korea, and Taiwan. Study designs included quasi-experimental (n = 26), cohort (n = 8), experimental (n = 9) and cross-sectional (n = 2) studies. Quasi-experimental studies involved interrupted times series with/without comparison (n = 9) and controlled/non-controlled before-after designs (n = 17). The study populations comprised patients in primary care clinics (n = 7), in outpatient medical care (n = 2), in units within hospitals or in hospitals (n = 29), consumers (n = 2), providers (n = 4) and purchasers (n = 1). The most common type of PPR were report cards (e.g. CABG report cards) (n = 12), reports (n = 13) and hospital comparisons websites (e.g. CMS Centres for Medicare & Medicaid Services) (n = 13). PPR quality indicators were predominantly reported at the hospital level

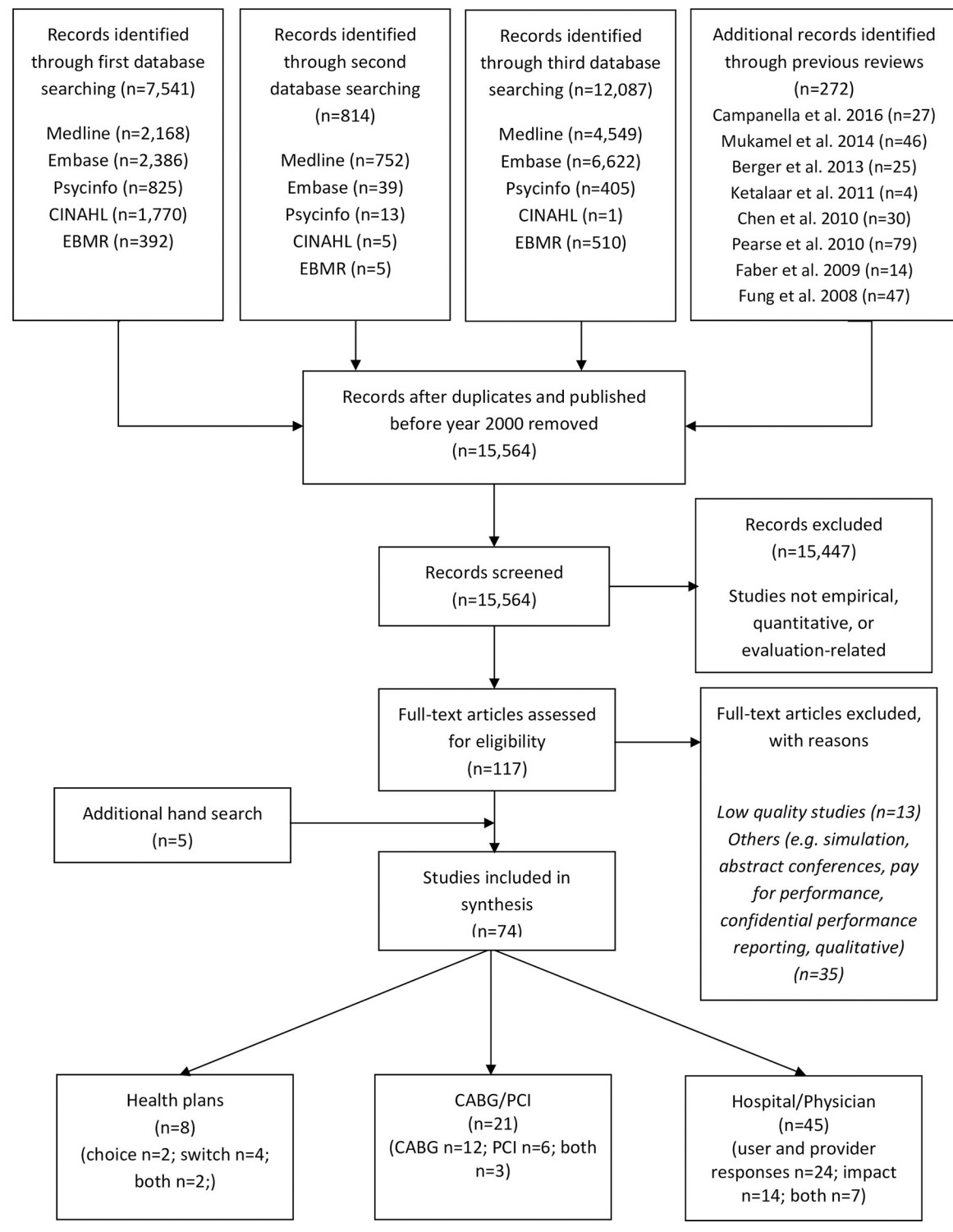

**Fig 1. Flow diagram for retrieval of articles.**

(n = 30), followed by individual physician/primary care clinics level (n = 14), and at the village level (n = 1). Nineteen studies examined mandatory PPR, 10 voluntary PPR, 15 compared PPR with no PPR and 1 compared mandatory PPR with voluntary PPR.

**Table 2. Characteristics and main findings of included studies.**

| Authors and year | Country (State/ Region/City) | Study design | Type of PPR | Findings* | Estimates |
|---|---|---|---|---|---|
| User and provider response (selection) | | | | | |
| Mukamel et al. 2002 [39] | USA (New York) | cohort study (retrospective) | Report cards[d] (CABG) | Positive effect | An increase of 1 standard deviation in excess RAMR leads to a decrease in the contract probability, p<0.01 |
| Mukamel et al. 2004 [40] | USA (New York) | quasi-experimental study (before-after study) | Report cards[d] (CABG) | Positive effect | Higher RAMR (i.e., lower quality) lowers the surgeon's odds of being selected by about 7% to 8%, p<0.01 |
| Werner et al. 2005 [43] | USA (New York) | quasi-experimental study (interrupted time series with comparison group) | Report cards[d] (CABG) | Negative effect | 2.0–3.4 percentage points between New York (PPR) and the comparison States (no PPR), p<0.01 |
| Epstein 2010 [44] | USA (Pennsylvania) | quasi-experimental study (controlled before-after study) | Report cards[d] (CABG) | No effect | Referral patterns to low-mortality (0.0 percentage points, SE = 0.8) or high-mortality cardiac surgeons (-0.3 percentage points, SE = 0.4) |
| Martino et al. 2012 [41] | USA (Michigan) | experimental study (randomised encouragement design) | Reports[d] (primary care quality) | Positive effect | Selected primary care physicians with higher scores on member satisfaction, β = 0.24, SE = 0.12, p = 0.04 |
| | | | | No effect | Overall clinical quality of primary care physician selected, β = 0.12, SE = 0.12, p = 0.33 |
| Ikkersheim & Koolman 2013 [42] | Netherlands (Eindhoven) | experimental study (randomised cluster trial) | Report cards[c] | Positive effect | For breast cancer, GPs refer 1% more to hospitals that score 1% point better on indicators for medical effectiveness, 95% CI (0.01 to 0.08), p = 0.01 |
| | | | | No effects | GPs referral patterns for cataract surgery, β = 0.01, 95% CI (-0.02 to 0.03), p = 0.74, and hip and knee replacement, β = -0.01, 95% CI (-0.03 to 0.01), p = 0.19 |
| Yu et al. 2018 [45] | Taiwan (national) | quasi-experimental study (before-after study) | Report cards[c] (Bureau of National Health Insurance) | Positive effect | Disadvantaged patients received care at excellent-performance hospitals post-program implementation, β = 0.05, SE = 0.01, p = 0.006 |
| Gourevitch et al. 2019 [46] | USA (national) | experimental study (randomised controlled trial) | Website[c] (The Leapfrog Group) | No effect | Proportion of women who selected hospitals with low caesarean delivery rates (7.0% control vs 6.8% intervention, p = 0.54) |
| Fabbri et al. 2019 [47] | India (Uttar Pradesh) | experimental study (factorial cluster-randomised controlled trial) | Report cards[f] | No effect | Proportion of women who had at least four antenatal care visits (provider vs non-provider: OR = 0.85, 95% CI (0.65 to 1.13), p = 0.264; community vs non-community: OR = 0.86, 95% CI (0.65 to 1.13), p = 0.276 |
| User and provider response (organisational quality improvement) | | | | | |
| Werner et al. 2008 [49] | USA (national) | cohort study (retrospective) | Website[c] (CMS Hospital Compare) | No effects | Hospitals with high percentages of Medicaid patients had smaller improvements in hospital performance than those with low percentages of Medicaid patients: composite scores for AMI absolute difference 1.5, 95% CI (0.2 to 2.9), p = 0.03; HF absolute difference 1.4, 95% CI (0.1 to 2.7), p = 0.04; pneumonia absolute difference 1.3, 95% CI (0.7 to 1.8), p<0.001 |
| Besley et al. 2009 [48] | UK (England, national) | quasi-experimental study (interrupted time series with comparison group) | Website[c] (NHS star rating) | Positive effect | The number of patients waiting between 9 and 12 months reduced by 67% |
| Bishop et al. 2012 [52] | USA (national) | cross-sectional study | Survey[c] (US National Ambulatory Medical Care) | Positive effect | Weight reduction counselling 10.0% (no PPR) vs 25.5% (PPR), p = 0.01 |
| | | | | No effects | Advising smokers and tobacco users to quit 24.1% (no PPR) vs 30.5% (PPR), p = 0.64; BMI screening 49.5% vs 49.6%, p = 0.85; urinalysis not performed at visit 93.1% vs 92.3%, p = 0.84; blood pressure management 45.7% vs 42.7%, p = 0.98; ACE-inhibitor or ARB therapy 45.4% vs 31.9%, p = 0.24; beta blocker therapy 55.3% vs 71.4%, p = 0.03; oral antiplatelet therapy 49.1% vs 47.8%, p = 0.89; beta blocker therapy 48.1% vs 53.4%, p = 0.48; no antibiotics for upper respiratory infection 46.0% vs 39.1%, p = 0.80; anticoagulation therapy in patients with atrial fibrillation 46.0% vs 39.1%, p = 0.30; bronchodilator therapy in patients with COPD 49.7% vs 55.6%, p = 0.23 |
| Leerapan 2011 [51] | USA (Minnesota) | quasi-experimental study (controlled before-after study) | Reports[d] (Minnesota Community Measurement Health Care Quality) | Positive effect | Average clinics with one lower percentile ranking had 0.2 higher percentage points of optimal diabetes care quality improvement on the next report, p<0.001 |

*(Continued)*

**Table 2.** (Continued)

| Authors and year | Country (State/Region/City) | Study design | Type of PPR | Findings* | Estimates |
|---|---|---|---|---|---|
| Jang et al. 2011 [50] | Korea (national) | quasi-experimental study (interrupted time series without comparison group) | Reports[c] (Korean Health Insurance Review & assessment service) | No effects | No effect for four repeated release of PPR except for the first which decreased the monthly national average caesarean section rate by 0.81%, $p<0.05$** |
| Renzi et al. 2012 [54] | Italy (Lazio) | quasi-experimental study (interrupted time series with comparison group) | Website[c] (Regional Outcome Evaluation Program) | Positive effects | AMI patients treated with PCI within 48 hours, RR = 1.31, $p<0.001$; hip fracture operations within 48 hours, RR = 1.34, $p<0.001$ |
| | | | | Negative effect | Primary caesarean deliveries, RR = 1.02, p = 0.012 |
| Smith et al. 2012 [53] | USA (Wisconsin) | cross-sectional | Reports[e] (Wisconsin Collaborative for Healthcare Quality) | Positive effect | Clinics focused on diabetes metrics more likely to implement at least one diabetes intervention, OR = 1.30, 95% CI (1.06 to1.60) |
| Wang et al. 2014 [56] | China (Hubei) | quasi-experimental study (interrupted-time series with comparison group) | Bulletin boards and brochures[d,e] | Positive effect | Reduction of 4% in injection prescribing rate, OR = 0.96, 95% CI (0.94 to 0.97), $p<0.001$ |
| Yang et al. 2014 [57] | China (Hubei) | experimental study (matched-pair cluster randomised trial) | Bulletin boards and brochures[d,e] | Positive effect | Oral antibiotics prescriptions 9.21 percentage points reduction, 95% CI (-17.36 to -1.07), p = 0.027 |
| | | | | No effects | IV injection prescriptions, 1.23 percentage points increase, 95% CI (-3.82 to 6.28), p = 0.633; infusion prescriptions, 1.37 percentage points increase, 95% CI (-3.93 to 6.67), p = 0.612 |
| Ukawa et al. 2014 [55] | Japan (national) | cohort study (retrospective) | Reports[c] (The quality indicator/improvement project) | Positive effects | 5.8 percentage points increase in composite score of five process measures |
| Kraska et al. 2016 [58] | Germany (national) | quasi-experimental study (controlled before-after study) | Reports[c] (German quality reports and external quality assurance) | Positive effects | Pacemaker implantation QI(I)-A compliant indication for bradycardia $\eta^2 = 0.22$, $p<0.001$; QI(I)-B compliant system selection for bradycardia $\eta^2 = 0.11$, $p<0.001$; Gynaecological surgery QI(P)-C Antibiotic prophylaxis in hysterectomy $\eta^2 = 0.07$, $p<0.001$; Obstetrics QI(P)-D presence of a paediatrician at premature births $\eta^2 = 0.04$, $p<0.001$, QI(P)-E antenatal corticosteroid therapy in premature birth with prepartum hospitalisation for at least two calendar days $\eta^2 = 0.13$, $p<0.001$; Coronary angiography QI(O)-F achieving the recanalization target in PCI with acute coronary syndrome with ST elevation up to 24h $\eta^2 = 0.02$, p = 0.002 |
| Lui et al. 2016 [60] | China (Hubei) | experimental study (cluster-randomised matched-pair trial) | Posters[d,e] | Positive effect | Combined antibiotics prescriptions, OR = 0.87, 95% CI (0.85 to 0.89), $p<0.001$ |
| | | | | Negative effects | Antibiotics prescriptions OR = 1.08, 95% CI (1.06 to 1.11), $p<0.001$; injections prescriptions OR = 1.25, 95% CI (1.23 to 1.28), $p<0.001$ |
| Tang et al. 2016 [61] | China (Hubei) | experimental study (cluster randomised matched-pair trial) | Posters[d,e], brochures, and reports | Positive effects | Antibiotics prescriptions for gastritis, 12.72% decrease, 95% CI (-16.59 to -8.85), $p<0.001$; combined antibiotics prescriptions for bronchitis, 3.79% decrease, 95% CI (-6.42 to -1.17), p = 0.005; injection prescriptions for gastritis), 10.59% decrease, 95% CI (-14.47 to -6.62), $p<0.001$; antibiotics injections prescriptions for gastritis, 10.73% decrease, 95% CI (-14.41 to -7.04) $p<0.001$ |
| | | | | Negative effects | Antibiotics prescriptions for hypertension 2.00% increase, 95% CI (0.53 to 3.47), p = 0.008; injection prescriptions for bronchitis 2.00% increase, 95% CI (0.43 to 3.56), p = 0.012 |
| | | | | No effects | Antibiotics prescriptions for bronchitis 0.02%, 95% CI (-0.9 to 0.09), p = 0.964; combined antibiotics prescriptions for gastritis -0.09%, 95% CI (-1.56 to 1.37), p = 0.898 and hypertension 0.44%, 95% CI (-0.04 to 0.91), p = 0.073; injection prescriptions for hypertension -0.97%, 95% CI (-3.37 to 1.43), p = 0.428; antibiotics injection prescriptions for bronchitis -0.07%, 95% CI (-2.02 to 1.87), p = 0.939 and hypertension -0.18%, 95% CI (-0.80 to 0.44), p = 0.569 |

(*Continued*)

**Table 2.** (Continued)

| Authors and year | Country (State/Region/City) | Study design | Type of PPR | Findings* | Estimates |
|---|---|---|---|---|---|
| Tang et al. 2017 [62] | China (Hubei) | experimental study (cluster randomised matched-pair trial) | Posters[d,e] brochures, and reports | Positive effects | Antibiotics prescription rate 2.82% reduction, 95% CI (-4.09 to -1.54), p<0.001; combined antibiotics prescription rate 3.81% reduction, 95% CI (-5.23 to -2.39), p<0.001 |
| | | | | No effect | Injection antibiotics prescription rate, 0.39% reduction, 95% CI (-1.75 to -0.97), p = 0.218 |
| Lind & Flug 2019 [59] | USA (national) | quasi-experimental study (controlled before-after study) | Website[c] (CMS Hospital Compare | Positive effects | Rate of MRI utilisation without prior conservative therapy decreased for outpatient hospitals in 2012 RR = 0.95, 95% CI (0.93 to 0.97), p<0.001, 2013 RR = 0.92, 95% CI (0.90 to 0.95), p<0.001, 2014 RR = 0.90, 95% CI (0.87 to 0.93), p<0.001, p<0.001 and outpatient clinics in 2010 RR = 0.98, 95% CI (0.97 to 0.98), p<0.001, 2011 RR = 0.96, 95% CI (0.94 to 0.97), p<0.001, 2012 RR = 0.94, 95% CI (0.92 to 0.97), p<0.001, 2013 RR = 0.91, 95% CI (0.89 to 0.94), p<0.001 and 2014 RR = 0.89, 95% CI (0.87 to 0.92), p<0.001 |
| | | | | No effects | Rate of MRI utilisation without prior conservative therapy for outpatient hospitals in 2010 RR = 1.00, 95% CI (0.98 to 1.03), p = 0.73, and 2011 RR = 0.98, 95% CI (0.95 to 1.00), p = 0.06 |
| Impact (clinical outcomes) | | | | | |
| Baker et al. 2002 [63] | USA (Ohio) | quasi-experimental study (interrupted time series without comparison group) | Report cards[c] (Cleveland Health Quality Choice) | Positive effects | In-hospital mortality absolute change for AMI -4.1, 95% CI (-6.4 to -1.5), p<0.005; CHF -3.7, 95% CI (-4.3 to -3.0), p<0.001; GIH -2.7, 95% CI (-3.6 to -1.4), p<0.001; COPD -2.1, 95% CI (-2.8 to -1.3), p<0.001; PNEU -4.8, 95% CI (-5.9 to -3.7), p<0.001; 30-day mortality absolute change for CHF -1.4, 95% CI (-2.5 to -0.1), p<0.05; COPD -1.6, 95% CI (-2.8 to 0.0), p<0.05 |
| | | | | Negative effects | Post discharge mortality absolute change for AMI 3.0, 95% CI (1.3 to 5.3), p<0.001; CHF 1.7, 95% CI (0.8 to 2.6), p<0.001; GIH 1.4, 95% CI (0.4 to 2.9), p<0.005; PNEU 2.3, 95% CI (1.4 to 3.5), p<0.001; STR 3.8, 95% CI (2.2 to 5.8), p<0.001; 30-day mortality absolute change for STR +4.3, 95% CI (1.8 to 7.1), p<0.001 |
| | | | | No effects | In-hospital mortality absolute change for STR -1.0, 95% CI (-2.6 to 0.9); post discharge mortality absolute change for COPD 0.7, 95% CI (-0.6 to 2.6); 30-day mortality absolute change for AMI -0.6, 95% CI -3.4 to 2.5), GIH -0.3, 95% CI (-1.9 to 1.8), PNEU -0.5, 95% CI (-2.1 to 1.3) |
| Clough et al. 2002 [64] | USA (Ohio) | cohort study (retrospective) | Report cards[c] | No effect | Mortality rate in Cleveland (-0.21% per 6 months, 95% CI (-0.27 to -0.15) vs rest of state (-0.18% per 6 months, 95% CI (-0.23 to -0.14), p = 0.35 |
| Baker et al. 2003 [65] | USA (Ohio) | quasi-experimental study (interrupted time series without comparison group) | Report cards[c] (Cleveland Health Quality Choice) | No effect | The absolute change in risk-adjusted 30-day mortality for "average" hospitals -0.5%, 95% CI (−1.8 to 1.0) for "below average" hospitals -0.8%, 95% CI (-2.9 to 1.8) and "worst" hospitals -0.4%, 95% CI (-2.3 to 1.7) |
| Caron et al. 2004 [66] | USA (Ohio) | cohort study (retrospective) | Report cards[c] (Cleveland Health Quality Choice) | Positive effects | Length of stay for AMI 93% improvement, CHF 100% improvement and stroke 100% improvement. Mortality for AMI 59% improvement, CHF 85% improvement and stroke 59% improvement. Primary caesarean delivery rate 76% improvement. VBAC delivery rate 67% improvement. Total caesarean delivery rate 67% improvement. |
| Hollenbeak et al. 2008 [67] | USA (Pennsylvania) | quasi-experimental study (controlled before-after study) | Reports[c] (Pennsylvania Health Care Cost Containment Council) | Positive effects | Mortality rates for AMI, CHF, haemorrhagic stroke, ischemic stroke, pneumonia, sepsis, range OR = 0.59–0.79 for Pennsylvania patients (PPR) vs. non-Pennsylvania patients (no/limited PPR) |

(*Continued*)

**Table 2.** (Continued)

| Authors and year | Country (State/Region/City) | Study design | Type of PPR | Findings[*] | Estimates |
|---|---|---|---|---|---|
| Noga et al. 2011 [71] | USA (Massachusetts) | quasi-experimental study (interrupted time series without comparison group) | Website[c] (Patients First) | Positive effects | Reduction in overall falls, β = -0.04, 95% CI (-0.06 to -0.02), p<0.001; reduction in overall falls with injury, β = -0.01, 95% CI (-0.02 to 0.00), p = 0.05 |
| Ryan et al. 2012 [68] | USA (national) | quasi-experimental study (interrupted time-series without comparison group) | Website[c] (CMS Hospital Compare) | Positive effect | HF 30-day mortality RR = 0.97, 95% CI (0.95 to 0.99) |
| | | | | No effects | Heart attack 30-day mortality, RR = 1.01, 95% CI (0.99 to 1.03); pneumonia 30-day mortality RR = 1.07, 95% CI (1.05 to 1.09) |
| Daneman et al. 2012 [69] | Canada (Ontario) | cohort study (retrospective) | Report[c] (the Ontario Ministry of Health and Long Term Care) | Positive effect | 26.7% reduction in clostridium difficile cases, 95% CI (21.4% to 31.6%) |
| Marsteller et al. 2014 [70] | USA (national) | quasi-experimental study (controlled before-after study) | Report[c] (On the CUSP:Stop BSI program) | Positive effects | Reduction in CLABSI rates in the first 6 months for voluntary PPR IRR = 0.73, 95% CI (0.56 to 0.94), p = 0.014; and >1 year mandatory PPR IRR = 0.83, 95% CI (0.70 to 0.99), p = 0.033 |
| DeVore et al. 2016 [73] | USA (national) | quasi-experimental study (controlled before-after study) | Website[c] (CMS Hospital Compare | No effects | 30-day readmission rates after PPR for MI -2.3%, 95% CI (-5.1 to 0.6), p = 0.72; HF -1.8%, 95% CI (-3.3 to -0.2), p = 0.19; pneumonia -2.0%, 95% CI (-4.1 to 0.2), p = 0.21; COPD -2.6%, 95% CI (-4.5 to -0.7), p = 0.11; diabetes 0.1%, 95% CI (-4.1 to 4.5), p = 0.58; 30-day mortality after PPR for MI -3.7%, 95% CI (-10.3 to 3.5), p = 0.75; HF 3.1%, 95% CI (-1.3 to 7.6), p = 0.15; pneumonia 2.6%, 95% CI (-2.6 to 8.2), p = 0.86; COPD -1.6%, 95% CI (-7.1 to 4.3), p = 0.54; diabetes -5.4%, 95% CI (-17.9 to 8.8), p = 0.54 |
| Joynt et al. 2016 [74] | USA (national) | quasi-experimental study (controlled before-after study) | Website[c] (CMS Hospital Compare | No effects | Less of a decline in 30-day mortality rates for PPR compared to no PPR. Quarterly change in mortality process-only reporting for AMI -0.28, CHF -0.21, pneumonia -0.21, all -0.23; quarterly change in mortality process and mortality reporting for AMI -0.13, CHF -0.06, pneumonia -0.10, all -0.09 |
| Martin 2019 [75] | USA (national) | quasi-experimental study (controlled before-after study) | Website[c] (Federal and state government mandated reporting -Reporting Hospital Quality Data for Annual Payment Update program) | Positive effects | State mandated PPR on the probability of dying while in hospital for HF -0.36 percentage points, p<0.001 and AMI -0.25 percentage points, p<0.01; State mandated PPR on length of stay for HF -0.24 days, p<0.001 and AMI -0.10 days, p<0.001 |
| | | | | Negative effects | Federal mandated PPR on probability of dying for HF 0.63 percentage points, p<0.001 and AMI 1.18 percentage points, p<0.001. Federal mandated PPR on length of stay for HF 13.61 percentage points, p<0.001 and AMI 22.48 percentage points, p<0.001. |
| Impact (patient experience) | | | | | |
| Ikkersheim & Koolman 2012 [72] | Netherlands (national) | quasi-experimental study (controlled before-after study) | Report cards[c] (Consumer Quality Index) | Positive effect | Improvement in hospital performance from 0.02, p<0.01 to 0.03, p<0.05 points |
| Mann et al. 2016 [76] | USA (national) | quasi-experimental study (before-after study) | Surveys[c] (HCAHPS) | Positive effect | 2.8% increase in patient satisfaction with physician communication p<0.001 |
| Both user and provider response (organisational quality improvement) and impact (clinical outcomes) | | | | | |
| Tu et al. 2009 [32] | Canada (Ontario) | experimental study (cluster randomised trial) | Report cards[c] (Enhanced Feedback for Effective Cardiac Treatment) | Positive effect | 30-day mortality rates for AMI 2.5% lower, 95% CI (0.1% to 4.9%), p = 0.045 |
| | | | | No effects | 30-day mortality rates for CHF -1.1%, 95% CI (-3.2% to 0.9%), p = 0.26; composite process-of-care indicator for AMI absolute change 1.5%, 95% CI (−2.2% to 5.1%), p = 0.43 and CHF absolute change 0.6%, 95% CI (−4.5% to 5.7%), p = 0.81 |

(*Continued*)

**Table 2.** (Continued)

| Authors and year | Country (State/ Region/City) | Study design | Type of PPR | Findings* | Estimates |
|---|---|---|---|---|---|
| Werner et al. 2010 [33] | USA (national) | quasi-experimental study (before-after study) | Website[c] (CMS Hospital Compare) | Positive effects | Composite performance measure for AMI 3.3 percentage points, p<0.001; HF 7.6 percentage points, p<0.001; pneumonia 8.8 percentage points, p<0.001; AMI mortality 0.6 percentage points, SE (-0.9 to -0.2), p<0.05; AMI length of stay 0.19 days, SE (-0.23 to -0.15), p<0.001; readmission rates 0.5 percentage points change, SE (-0.9 to -0.2), p<0.01; HF readmission rates, 0.2 percentage points change, SE (-0.3 to -0.1), p<0.001 |
| | | | | Negative effect | Pneumonia length of stay 0.13 days, SE (0.1 to 0.16), p<0.001 |
| | | | | No effects | HF mortality 0.04 percentage point change, SE (-0.04 to 0.1); HF length of stay 0.01 days, SE (-0.02 to 0.02); pneumonia mortality 0.2 percentage point change, SE (-0.40 to -0.10); pneumonia readmission rates 0.1 percentage point change, SE (-0.3 to 0.1) |
| Reineck et al.2015 [35] | USA (California) | cohort study (retrospective) | Reports[c] (the California Health Care Foundation) | Positive effect | Post-acute care use OR = 0.94, 95% CI (0.91 to 0.96), p<0.001 |
| | | | | Negative effect | Transfer to another acute care hospital OR = 1.43, 95% CI (1.33 to 1.53), p<0.001 |
| | | | | No effects | In-hospital mortality OR = 0.99, 95% CI (0.95 to 1.03), p = 0.72; 30-day mortality OR = 0.99, 95% CI (0.96–1.02), p = 0.55 |
| Yamana et al. 2018 [38] | Japan (national) | quasi-experimental study (controlled before-after study) | Reports[c] (Ministry of Health, Labour and Welfare) | No effects | Risk-adjusted in-hospital mortality OR = 0.98, 95% CI (0.81 to 1.17), p = 0.789; aspirin within 2 days of admission OR = 1.03, 95% CI (0.81 to 1.30), p = 0.826 |
| Selvaratnam et al. 2020 [37] | Australia (Victoria) | cohort study (retrospective) | Report[c] (Safer Care Victoria's Perinatal Services Performance Indicators) | Positive effects | Reduction per quarter in percentage of severe small for gestational age babies undelivered by 40 weeks of gestation from 0.13% to 0.51%, p<0.001; decrease mortality rate for severely small for gestational age babies 3.3 per 1000 births, p = 0.01 |
| Dahlke et al. 2014[a] [34] | USA (national) | quasi-experimental study (controlled before-after study) | Website[c] (CMS Hospital Compare) | Positive effects | Accidental puncture or laceration OR = 2.11, 95% CI (1.04 to4.30), p<0.05; heart attack patient given aspirin at arrival OR = 0.26, 95% CI (0.10–0.64), p<0.05; heart attack patient given aspirin at discharge OR = 0.38, 95% CI (0.16–0.87), p<0.05; definitely recommending the hospital to friends and family OR = 0.28, 95% CI (0.11 to0.71), p<0.05 |
| | | | | No effects | Mortality for heart attack OR = 0.75, 95% CI (0.33 to 1.72); HF OR = 0.91, 95% CI (0.91 to 1.43); pneumonia OR = 0.69, 95% CI (0.30 to 1.58); readmission for heart attack OR = 0.51, 95% CI (0.19 to 1.43), HF OR = 0.91, 95% CI (0.39 to 2.14); pneumonia OR = 0.92, 95% CI (0.39 to 2.17); HCAHPS clean bathrooms OR = 0.58, 95% CI (0.26 to 1.28); nurses communication OR = 0.53, 95% CI (0.24–1.19); doctor communication OR = 0.86, 95% CI (0.39 to 1.91); help from staff OR = 0.95, 95% CI (0.44 to 2.06); pain controlled OR = 0.97, 95% CI (0.46 to 2.08); medications explanation OR = 0.54, 95% CI (0.23 to 1.23); home recovery information OR = 0.50, 95% CI (0.20 to 1.21); hospital quality OR = 0.52, 95% CI (0.23 to 1.16); quiet hospital rooms OR = 0.59, 95% CI (0.28 to 1.24); process measures** |

*(Continued)*

**Table 2.** (Continued)

| Authors and year | Country (State/Region/City) | Study design | Type of PPR | Findings[*] | Estimates |
|---|---|---|---|---|---|
| Vallance et al. 2018[b] [36] | UK (England, national) | quasi-experimental study (controlled before-after study) | Website[d] (NHS Choices and Association of Coloproctology of Great Britain and Ireland) | Positive effect | 90-day mortality for major colorectal resection decreased from 2.8% to 2.1%, p = 0.03 |
| | | | | No effect | Physician risk aversion measured as predicted 90 days mortality based on characteristics of patients and tumours (2.7% before PPR and 2.7% after PPR, p = 0.3) |

[*]No effect refers to no statistically significant effect of PPR

[**]See Table 3 in Jang et al. [50] for ARIMA models estimates and supplemental Table 2 in Dahlke et al. [34] for process measures estimates.

[a] Organisational quality improvement, Clinical outcomes, Patient experience

[b] Selection, Clinical outcomes

[c] Level of reporting hospital

[d] Level of reporting physician

[e] Level of reporting clinic

[f] Level of reporting village.

AMI acute myocardial infarction; BMI body mass index; CABG coronary artery bypass graft; CHF congestive heart failure; CI confidence intervals; CLABSI central line-associated bloodstream infection; CLABSI central-line associated bloodstream infection; CMS Centers for Medicare and Medicaid Services; COPD chronic obstructive pulmonary disease; GIH gastrointestinal haemorrhage; GPs general practitioners; HCAHPS Hospital Consumer Assessment of Healthcare Providers and Systems; HF heart failure; IRR incidence rate ratio; MRI magnetic resonance imaging; OR odds ratio; PCI percutaneous coronary intervention; PNEU pneumonia; PPR public performance reporting; RAMR risk adjusted mortality rate; RR risk ratio; SE standard errors; STR stroke.

**User and provider responses.** *Selection of patients, physicians and hospitals.* Eight studies examined the effects of PPR on the selection of physicians and hospitals by patients, consumers, healthcare purchasers and providers [39–42,44–47]. Two studies examined if there were detrimental effects of PPR on adverse selection of patients by physicians [36,43]. Yu et al. [45], Mukamel et al. [40], and Martino et al. [41] reported positive effects of PPR on the selection of hospitals, cardiac surgeons, and primary healthcare physicians by patients/consumers. Gouveritch et al. [46] reported no effects of PPR on the selection of hospitals with lower caesarean delivery rates by pregnant women. Similarly, Fabbri et al. [47] reported no effects of PPR on the proportion of women who received maternal and neonatal health care services. Epstein et al. [44] reported no effect of PPR on the selection of cardiac surgeons by physicians when referring patients. In contrast, Ikkersheim and Kohlmann [42] reported that publicly reporting quality indicators and patient experiences positively influenced general practitioners' choice of hospital when referring patients. Mukamel et al. [39] reported that cardiac surgeons with low risk-adjusted mortality rates (RAMR) were more likely to be contracted by managed care organisations than those with high RAMR. Werner et al. [43] reported that publicly reporting individual's surgeon performance resulted in an increase in racial and ethnic disparities in CABG use in New York compared to other States without PPR. Surgeons avoided operating on high-risk patients. In contrast, Vallance et al. [36] found no evidence that publicly reporting individual's surgeon 90-day postoperative mortality in elective colorectal cancer surgery has led to risk averse behaviours in England. The proportion of patients undergoing elective colorectal cancer surgery before and after the introduction of PPR remained the same. In summary, half of the studies reported positive effects of PPR, with one a negative effect and the rest no effect. These findings suggest moderate level of evidence for PPR and selection of patients, physicians and hospitals.

*Organisational quality improvement.* Twenty-one studies examined the effects of PPR on quality improvement activities in primary care clinics (n = 7), outpatient medical care (n = 2) and hospitals (n = 12). Among primary care clinics, Smith et al. [53] found that publicly

reporting diabetes care performance led to an increase in the number of diabetes quality improvement interventions implemented. Interventions included patient (e.g. education), provider (e.g. performance feedback) or system-directed (e.g. guidelines) interventions. Similarly, Leerapan [51] found that publicly reporting the rankings of primary care clinics improved the quality of diabetes care provided, in particular among lower rank clinics. Wang et al. [56], Yang et al. [57], and Lui et al. [60] found that publicly reporting both primary care clinics and individual physicians' prescription rates reduced their prescription rates of antibiotics and injections, thereby potentially reducing medication overuse. Using the same data derived from Lui et al.'s study [60], Tang et al. stratified the analysis by health conditions [61] and physician's prescribing performance level [62]. The effect of PPR varied by health conditions, with a reduction in antibiotics and injections prescriptions for patients with gastritis compared to patients with bronchitis or hypertension [61]. There was a decrease in the rate of antibiotics prescriptions following PPR across all physician's prescribing performance level, with the effect largely attributed to average and high antibiotic prescribers [62].

Among outpatient medical care, Lind et al. [59] found that publicly reporting imaging efficiency indicator resulted in an improvement in the appropriate use of conservative therapy and imaging among patients with low back pain. In contrast, Bishop et al. [52] found no associations between PPR of practice measures and 12 quality indicators related to preventative care, diabetes mellitus, heart failure and coronary artery disease, except for one preventative care measure—weight reduction counselling for overweight patients (see S5 Appendix for full list of measures).

Among hospitals, Besley et al. [48] reported that mandatory PPR with targets and sanctions (naming and shaming) in England reduced waiting times for elective care, compared to Wales which did not implement these initiatives. However, there was some evidence of moving patients around to meet targets in England. Similarly, Reinecke et al. [35] found that PPR in California reduced post-acute care use but increased acute care hospital transfer rates among intensive care unit (ICU) patients compared to other States without PPR. Werner et al. [33] reported an improvement in all process measures for acute myocardial infarction, heart failure and pneumonia following PPR, particularly in hospitals with low baseline performance (see S5 Appendix). Similarly, both Kraska et al. [58] and Selvaratnam et al. [37] found an improvement in care delivery processes following PPR of clinical quality indicators (see S5 Appendix). Renzi et al. [54] and Ukawa et al. [55] reported that hospitals who participated in PPR had better performance in several process measures than hospitals who did not (see S5 Appendix). Specifically, Renzi et al. [54] found that PPR resulted in an increase in PCI and hip fracture operations within 48 hours but minimal impact on caesarean section rates. Jang et al. [50] also reported no impact of PPR on caesarean section rates beyond the first release of PPR. Werner et al. [49], Tu et al. [32] and Dahlke et al. [34], and Yamana et al. [38] reported limited or no impact of PPR on a number of process measures related to heart attack and failure, pneumonia and surgical care (see S5 Appendix). In particularly, Werner et al. [49] noted that hospitals with high percentages of Medicaid patients had smaller improvements in hospital performance than those with low percentages of Medicaid patients. In summary, all studies reported positive effects of PPR for primary care (although the findings of three studies appeared to be derived from one RCT [60–62], and half of the studies reported positive effects of PPR for outpatients and hospitals. These findings suggest strong and moderate level of evidence for PPR and quality improvement activities in primary care and hospitals, respectively but inconclusive evidence for outpatients given the low number of studies.

**Impact of PPR on quality of care.** *Improvement in clinical outcomes.* Nineteen studies examined the impact of PPR on clinical outcomes. The most common clinical outcome indicator was mortality (n = 16) [32–38,63–68,73–75]. Seven studies reported no effects of PPR on

mortality in general inpatient care [34,38,64,65,73,74] and intensive care [35]. In contrast, six studies reported that PPR reduced mortality in general inpatient care [32,33,36,66,67] and perinatal care [37]. Three studies showed mixed effects of PPR on mortality depending on the health conditions [63,68] and the level of reporting (State or Federal) [75].

Other clinical outcome indicators included readmission rates, infection rates and falls. Werner et al. [33] reported that PPR was associated with a decline in 30-day readmission rates among patients with AMI, heart failure or pneumonia, whilst both Dhalke et al. [34] and DeVore et al. [73] reported no PPR effects. The conflicting results may be due to the different time periods investigated. Both Danemann et al. [69] and Marsteller et al. [70] reported that mandatory PPR of hospital-acquired infection rates reduced infection rates in hospitals. Similarly, Noga et al. [71] found that hospitals who volunteered to publicly report their patients' falls with and without injuries had a decrease in patients' falls.

*Improvement in patient experience.* Three studies examined the impact of PPR on non-clinical outcomes such as patient experience. Mann et al. [76] reported that patient satisfaction with physician communication increased following mandatory public reporting of patients' perception of hospital care survey, with the largest improvement occurring among hospitals in the lowest quartile of satisfaction scores. Ikkersheim et al. [72] found that hospitals that were 'forced' to publicly publish their Consumer Quality Index results by health plans insurers had better patient experiences than those who did not. In contrast, Dahlke et al. [34] reported mostly no effects of PPR on patient experiences (with the exception of "definitely recommending the hospital") between hospitals who volunteer to publicly publish their performance and those who do not. In summary, the majority of the studies reported positive effects of PPR on clinical outcomes including mortality (six of 16) and readmission rates, infection rates and falls (four of six), and patient experience (two of three). These findings suggest moderate level of evidence for PPR and clinical outcomes and some evidence for PPR and patient experience, albeit the low number of studies.

## Discussion

This systematic review summarises the evidence on the mechanisms and impacts of PPR on physicians and hospitals' performance. Among user and provider behavioural responses studies, five of 10 studies reported a positive effect of PPR on the selection of healthcare providers by patients, physicians and purchasers; 15 of 21 studies reported positive effects of PPR on quality improvement activities in primary care clinics and hospitals. Among impacts of PPR studies, 10 of 19 studies reported positive effects of PPR on clinical outcomes and two of three studies on patient experience. Only one study reported a negative effect of PPR on the selection of patients by healthcare providers.

Previous PPR reviews have yielded conflicting results; early reviews demonstrated associations between PPR and improvement in processes of care and clinical outcomes [13,15,23], although follow-up reviews showed limited associations [16,22]. There were also inconsistent associations between PPR and selection of healthcare providers [13,15,16,19,22]. Given that PPR may exert different effects across healthcare settings and health conditions, our reviews extend these results by considering the effects of PPR by procedures for specific condition [21], consumer choice pertaining to health plans [20] and physicians and hospitals performance focusing on the mechanisms and impacts of PPR, in which the findings are reported here. Consistent with previous reviews [13,15,18,19,23], we found that PPR stimulate quality improvement activities and improve clinical outcomes including mortality.

The majority of studies showed that PPR positively influenced the selection of healthcare providers (i.e. individual physician, hospital) by patients, providers, and purchasers. This is

consistent with the findings of reviews conducted by Chen et al. [15] and Vukovic et al. [19] but not others [13,22]. The discrepancy between reviews likely reflects the healthcare choices consumers and healthcare providers are asked to make, as some reviews incorporated selection of healthcare providers, health plans and nursing homes together, and used hospital's surgical volume and market share as measures of selection. All studies included in our review focused on actual consumer choice behaviour in the hospital and physician sector of health services. The findings related to the selection of health plans [20] and market share associated with CABG/PCI [21] are reported separately. Although the findings suggest that consumers are aware of PPR data, understand it and use it to make an informed choice, the results warrant cautious interpretation given the small number of studies across consumer types. Across the studies, quality indicators in the report cards included a mix of process and outcome measures for a specific health condition or procedure reported at the individual physician or hospital level. Previous studies have demonstrated that patients are interested in interpersonal aspect of care indicators (e.g. patient experience and satisfaction) reported at the individual physician level [77–79]; whereas providers and purchasers considered processes and outcomes measures (e.g. surgical complications and mortality) to be important indicators that should be publicly reported [80,81]. Consumer-focused frameworks and best practice guidelines have also been developed for presenting, promoting and disseminating PPR data to improve their comprehensibility and usability [24,82].

The effects of PPR on quality improvement activities appeared to be dependent on the healthcare setting, type of process indicators publicly reported and the clinical areas it is reported for. Among primary care clinics, publicly reporting individual physician and clinic care performance and ranking their performance resulted in positive behavioural changes [51,53,56,57,60–62]. This suggests that PPR improves performance via a feedback loop. Similar positive effects of PPR on quality improvement activities were observed in hospitals, however the effects varied across clinical areas [33,35,37,48,54,55,58]. The differential effects of PPR across clinical areas may be related to the type of process indicators reported, as some may be more amenable to behavioural change. For example, the cardiac and orthopaedic process measures focus on the proportion of patients treated with a surgical procedure within recommended time or medication at admission or discharge from hospital which may allow for timely targeted behavioural change [33,54,55]. In comparison, obstetrics and respiratory process measures such as the proportion of women with primary caesarean and pneumococcal vaccination quantify the measures but provide no guidance on how to improve caesarean and pneumococcal vaccination rates [34,46,49,50,54]. Given there can be substantial variation in quality of care across the different departments of a hospital, implementing and tracking relevant evidence-based process metrics for individual clinical areas are necessary to drive quality improvement and reduce variation in care delivery.

Although process measures may drive quality improvement activities, it remains unclear whether they lead to successful clinical outcomes. This is likely to be dependent on whether the process measures are evidence-based or not. Evidence-based process measures generally reflect accepted recommendations for clinical practice [83]. Furthermore, strict adherence to process measures, in the form of 'targets', may be detrimental to clinical outcomes and lead to unintended consequences such as 'gaming' (i.e. shuffling of patients to meet targets), 'cream skimming' (i.e. admitting healthier patients), and risk aversion. Two of three studies in our review found evidence of gaming associated with targets and sanctions [48], and risk aversion behaviours by surgeons [43]. In support, previous reviews have reported similar unintended and negative consequences of PPR on patients and healthcare providers [84–86]. To mitigate the unintended consequences of PPR, Marshall et al. [87] suggested a broader assessment of performance beyond process measures that reflect the effectiveness and quality of care, such as

clinical outcomes, patient experience and satisfaction measures. Custers et al. [88] proposed using incentive structure (e.g. payments for targets or penalties for gaming) alongside PPR to influence healthcare providers' attitudes. In support, a previous US study found that hospitals subject to both PPR and financial incentives improved quality more than hospitals engaged only in PPR [89].

The majority of studies showed positive impact of PPR on the improvement of clinical outcomes, in particular mortality. Mortality is considered an objective endpoint that is easily measurable and understandable by the public [90]. Despite this, it is unclear what quality improvement activities individual physicians and hospitals undertook to improve their mortality rates as using clinical outcome measures alone can make it difficult to identify a specific gap in care. As such, measurement of processes rather than outcomes of clinical care has been proposed as a more reliable and useful measure for quality improvement purposes [91]. However, as discussed above using solely process measures may be more susceptible to unintended consequences. Having a balance of relevant process and outcomes measures is preferable to minimise negative consequences [87].

Other clinical outcomes such as functioning (i.e. the lived experience of health) [92], health-related quality of life, patient-reported outcomes and experiences were rarely investigated. In our review, only three studies [34,72,76] examined patient experience and two found positive effects of PPR on patient experience [72,76]. Previous reviews reported positive effects of PPR on patient experience, but this was limited to one or two studies involving hospital reimbursements linked to patient experience scores [19,27]. We did not include pay for performance studies in our review as these effects could not be disaggregated from PPR. Given the growth in patient-centred care, many healthcare systems such as the US and UK are publishing inpatient hospital experience [3]. The impact of publishing them appeared to be positive to date but further empirical studies are warranted given the low number of studies.

Additional factors that could have an influence on the impact of PPR on quality improvement activities and clinical outcomes include the structural characteristics and culture of the hospitals. Two studies in our review examined hospital structural characteristics [34,55]. Both Ukawa et al. [55] and Dahlke et al. [34] found that hospitals which voluntarily participated in PPR had higher baseline performances. Aside from this, there were few hospital structural characteristics differences between hospitals that voluntary participated in PPR and those that did not. This suggests that past hospital's performance may influence the initial decision to voluntary participate in PPR but may not be the sole driver. Previous studies had shown that hospitals with strong quality and safety culture were more likely to engage in quality improvement activities and tended to have higher publicly reported hospital rating scores [93,94]. A qualitative study of hospital Medical Directors' views identified strong leadership and organisational cultures that encourage continuous quality improvement and learnings as important for open and transparent reporting of performance data [95].

## Implications

Public reporting of hospital performance data has become a common health policy tool to inform consumer healthcare choice, as well as stimulate and maintain quality improvement in clinical practice. When devising a PPR strategy, health policy makers must identify who the intended audience (i.e. consumers, providers, purchasers) and the objectives (i.e. selection, quality improvement, transparency/accountability) of PPR are to increase its effectiveness [96].

For consumers, PPR can facilitate choice in selecting a physician or a hospital that appeared to have better outcomes if 1) the indicators are disseminated through the appropriate channel

to increase reach and awareness and 2) the indicators reported meet their decision-making needs. Meeting these prerequisites for PPR to be effective are dependent on consumers' characteristics that influence information-seeking and decision-making behaviours such as their health condition (urgency of care), level of education and health literacy. As such, health policy makers responsible for the development and dissemination of PPR must ensure that the indicators publicly reported are relevant and meaningful, publicised and published in accessible formats, easily understood and made readily available [97].

For providers, PPR data can be used to assess the performance of their organisation or their individual staff member when implementing quality improvement initiatives. PPR is a complex improvement intervention of which the actual 'change' mechanism that translate PPR into quality improvement initiatives is not yet well understood. This is key to understanding which quality improvement initiatives work under what condition and will ensure learnings are transferred and adopted across healthcare settings. However, PPR is only one strategy for the continuous improvement of hospital quality and safety. The US and several European countries are increasingly moving toward pay-for performance as a quality improvement strategy [98,99].

Finally, an assessment of whether PPR will be successful needs to consider the healthcare delivery system in which PPR operates. Most of the literature included in this review was derived from the experience of PPR in the US, which may not be applicable to other countries. The US healthcare system is a private insurance system that promotes healthcare choice and market competition. In contrast, the UK and Australia have universal health care systems with dual public and private healthcare sectors, where voluntary private insurance reduces access fees. Although citizens have free access to the universal public system, they may have fewer choice in their medical specialist and place of care than the private system. Furthermore, in these countries and others European countries, general practitioners (GPs) are generally gatekeepers to secondary care with patients requiring their referral for access [100]. There have been few studies examining whether PPR of hospital data influences GPs referral behaviour [80,101,102]. Given the growth of PPR outside of the US, health policy makers must consider other potential users of PPR beyond patients such as the intermediate role that GPs play in connecting patients with hospitals.

## Strengths and limitations

Whilst the search was extensive and included a wide range of relevant electronic databases, it did not include studies in languages other than English, grey literature, or qualitative studies. Studies that did not explicitly describe their research design may have also been missed. However, to minimise this risk, the search strategy was conducted with the assistance of a librarian and a second search was conducted to include non-standard epidemiological terminology. Although some risk of bias can be drawn from the methodological quality summary scores, they are a subjective judgment and have been previously criticised for ascribing equal weight to each of the nominated criteria [103]. Given that there is a lack of consensus on which is the best tools to assess the methodological quality of observational studies, the NOS was considered to be appropriate. We acknowledged that the methodological quality of the included studies should be interpreted with caution. We attempted to disentangle the effects of PPR by reporting the results by mechanisms and impacts across a range of users, healthcare settings and clinical areas. However, the small number of studies across users and clinical areas limit the strength of the evidence and the results warrant cautious interpretation. Due to the high level of heterogeneity in settings and outcomes between the studies, it was not possible to pool the results and conduct a meta-analysis. Finally, the literature has overwhelmingly been derived from one country and one health system (US).

In summary, we have found moderate evidence that PPR informed choice of healthcare providers, increased quality improvement activities, improved clinical outcomes, and patient experience (albeit the low number of studies), with some variations across healthcare settings and conditions. Ultimately, for PPR to be effective, the design and implementation of PPR must considered the perspectives and needs of different users, as well as the values and goals of the healthcare system in which PPR operates. There is a need to account for systems-level barriers such as the structural characteristics and culture of the hospitals that could influence the uptake of PPR. Accounting for these contextual elements have the potential to substantially increase the impact of PPR in meeting its objectives of increased transparency and accountability within the healthcare system, informing healthcare decision-making and improving the quality of healthcare services.

## Supporting information

**S1 Checklist.**
(DOC)

**S1 Appendix. Medline search strategy.**
(DOCX)

**S2 Appendix. Screening guide.**
(DOCX)

**S3 Appendix. Risk of bias assessment.**
(DOCX)

**S4 Appendix. Data extraction for studies considered to be of low methodological quality following risk of bias assessment.**
(DOCX)

**S5 Appendix. Quality indicators reported in the studies.**
(DOCX)

## Acknowledgments

The authors thank Dr Stuart McLennan who conducted the first search, Dr Angela Nicholas and Andrea Timothy for screening the titles and abstracts from the first search, Angela Zhang for conducting risk of bias assessment and data extraction of studies from the third search as a second assessor, and Jim Berryman for assisting in the search strategies.

## Author Contributions

**Conceptualization:** David Dunt, Margaret Kelaher.

**Data curation:** Khic-Houy Prang, Hana Sabanovic.

**Formal analysis:** Khic-Houy Prang, Roxanne Maritz.

**Funding acquisition:** David Dunt, Margaret Kelaher.

**Supervision:** Margaret Kelaher.

**Writing – original draft:** Khic-Houy Prang, Roxanne Maritz.

**Writing – review & editing:** Hana Sabanovic, David Dunt, Margaret Kelaher.

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
