## [Decision Letter · Decision Letter 0]

15 Apr 2020

PONE-D-20-05627

Mechanisms and impact of public reporting on physicians and hospitals' performance: A systematic review (2000-2016)

PLOS ONE

Dear Dr Prang,

Thank you for submitting your manuscript to PLOS ONE. After careful consideration, we feel that it has merit but does not fully meet PLOS ONE’s publication criteria as it currently stands. Therefore, we invite you to submit a revised version of the manuscript that addresses the points raised during the review process.

Please address all issues raised by the reviewer, including the one referred to the search end date. The current date,  April 2015, is not acceptable.

We would appreciate receiving your revised manuscript by May 30 2020 11:59PM. To enhance the reproducibility of your results, we recommend that if applicable you deposit your laboratory protocols in protocols.io, where a protocol can be assigned its own identifier (DOI) such that it can be cited independently in the future. For instructions see: http://journals.plos.org/plosone/s/submission-guidelines#loc-laboratory-protocols

We look forward to receiving your revised manuscript.

Kind regards,

Lamberto Manzoli, M.D., M.P.H.

Academic Editor

PLOS ONE

Journal Requirements:

2. Please ensure that your search is up to date, in order to allow the inclusion of studies published within the past 12 months.

Reviewers' comments:

Reviewer's Responses to Questions

**Comments to the Author**

1. Is the manuscript technically sound, and do the data support the conclusions?

Reviewer #1: Yes

2. Has the statistical analysis been performed appropriately and rigorously? 

Reviewer #1: N/A

3. Have the authors made all data underlying the findings in their manuscript fully available?

Reviewer #1: Yes

4. Is the manuscript presented in an intelligible fashion and written in standard English?

Reviewer #1: Yes

5. Review Comments to the Author

Reviewer #1: Recent paper by Prang et al. named “Mechanisms and impact of public reporting on physicians and hospitals' performance: A systematic review (2000-2016)” tried to summarize the impact of PPR on physicians and hospitals’ performance. The overall idea behind this research is legitimate with strong worldwide interest in the topic but the general impression is that the manuscript could be improved by better specifying some crucial elements. Paper itself is without a strong conclusion, introducing the topic with the statement “However, previous studies have demonstrated inconsistent effects of PPR, potentially due to the various PPR characteristics examined.” and concluding in the same manner: “There was limited and inconclusive evidence to demonstrate a relationship between PPR and patient experience.” This should be modified.

Overall, the introduction needs to better explain the problematic behind the need to have this kind of research especially in respect to the complexity of the topic, should be discussed furthermore with stronger arguments for conducting this kind of research. Further, there any several already available (and much more structured) reviews published and authors should emphasize why conducting other, especially since conclusions remained more-less the same as in previously published reviews on the topic. Also, considering the study design here, it could easily be an umbrella-review? Could authors explain why they didn’t opt for this kind of review since again there are already several reviews available which they considered for the additional reference search?

In the introduction author mention two pathways through with PPR can improve quality of care, when there are actually three pathways suggested in the literature, third is the reputation pathway introduced by Hibbard et al (2005). This should be modified.

Methods: Why there was a time-limit year 2000? Also, search should be updated after 16th April 2015 until 2020 since several studies have been published and this might give different conclusions. Or at least reasons for not doing so should be documented. Inclusion and exclusion criteria are well defined and PRISMA guide was followed.

Results section: should be synthesised a bit more, following the common pattern and not just retelling findings from the tables.

Discussion: Most of the finding have already been described in the previous reviews, with perhaps difference of those reported previously that here some healthcare settings and types of providers were excluded (!?). Finally, the main question that remains unanswered – what should one do with these findings? This should be clearly summarized - what this research adds to the previous already available? Also, policy implications and/or further recommendations are lacking here, these should be added.

6. PLOS authors have the option to publish the peer review history of their article (what does this mean?). If published, this will include your full peer review and any attached files.

Reviewer #1: No

---

## [Author Response · Author response to Decision Letter 0]

17 Sep 2020

Please see our response to reviewers file.

---

## [Decision Letter · Decision Letter 1]

14 Oct 2020

PONE-D-20-05627R1

Mechanisms and impact of public reporting on physicians and hospitals' performance: A systematic review (2000-2020)

PLOS ONE

Dear Dr. Prang,

Thank you for submitting your manuscript to PLOS ONE. After careful consideration, we feel that it has merit but does not fully meet PLOS ONE’s publication criteria as it currently stands. Therefore, we invite you to submit a revised version of the manuscript that addresses the points raised during the review process.

Please address all issues raised by the reviewer, with a special focus on the exclusion of studies because of the low quality. This is a crucial point: in general, no study should be excluded because of an evaluation based on a biased tool such as the Newcastle Ottawa scale. Thus, all studies should be discussed, and their limitations mentioned and considered. Also, the report of the findings of the individual studies is too scarce, and quantitative estimates of effects should be reported. This will also help understanding whether it was really impossible to perform a meta-analysis, of it might have been performed on some specific outcomes.

We look forward to receiving your revised manuscript.

Kind regards,

Lamberto Manzoli, M.D., M.P.H.

Academic Editor

PLOS ONE

Reviewers' comments:

Reviewer's Responses to Questions

**Comments to the Author**

1. If the authors have adequately addressed your comments raised in a previous round of review and you feel that this manuscript is now acceptable for publication, you may indicate that here to bypass the “Comments to the Author” section, enter your conflict of interest statement in the “Confidential to Editor” section, and submit your "Accept" recommendation.

Reviewer #1: (No Response)

2. Is the manuscript technically sound, and do the data support the conclusions?

Reviewer #1: Yes

3. Has the statistical analysis been performed appropriately and rigorously? 

Reviewer #1: N/A

4. Have the authors made all data underlying the findings in their manuscript fully available?

Reviewer #1: Yes

5. Is the manuscript presented in an intelligible fashion and written in standard English?

Reviewer #1: Yes

6. Review Comments to the Author

Reviewer #1: Compliments to the authors for addressing previously raised issues and for these additional efforts when updating the literature search which all together significantly improved quality of the research. I’m very much pleased to see that the number of included studies increased 50% which further reflects the importance of this topic, where in just 4 years for which the search was now extended the authors found 15 new eligible studies.

Here are some minor, mostly technical, modifications that should be amended before final decision:

Introduction was enriched with several good explanations bringing it to more… also, adding table 1 with classification of PR by mechanisms and audience helps reader having a better picture of the topic.

Even though during this update of the literature search you have probably included all the available eligible studies covered by the following reviews (published in years after 2016), but for the sake of comprehensiveness these should also be acknowledged (after double checking it) in the search diagram (Fig.2) in the part "additional records identified through previous reviews". You have already cited most of these reviews throughout the discussion so it's unclear why this is not presented also in the diagram.

Several studies with assigned low quality have been excluded in the step “Full-text articles assessed for eligibility” which is not in line with the exclusion criteria reported in the methods. Usually, evaluation of the methodological quality of studies comes after the final selection of the eligible studies since here it’s very easy to slip into biased conclusion and some potentially valuable conclusions, based on quality data, might come even from studies with low(er) methodological quality. Finally, I would encourage authors to keep these studies, perhaps in a separate supplementary table so the readers could have a general (and full) idea of the available evidence, while not considering findings from these low quality studies when making the main conclusion here. In case the authors decide to go with the current setting, the low quality of studies should be clearly stated as the exclusion criteria.

I wasn’t able to find the supplementary table with QA of the included studies using Newcastle-Ottawa Scale nor the Cochrane Collaboration’s tool for assessing risk of bias? If not placed somewhere where I might’ve missed it, this should be added in the supplementary tables. Also, in the text please always refer to the “methodological” quality in sentences when describing quality of the studies.

Please consider placing Figure 1 and Figure 2 into one, the reader doesn’t need to be bothered with two diagrams since this is an issue of this study design of this study (search of the literature repeated several times).

Description of the studies – lines 187/188, please try rephrasing when summing up the studies “by two from Canada and Japan…. by one from India, Germany…” instead if listing one by one, in general would be better to synthesize this descriptive part for the reader.

Table 2 could benefit from some restyling. First, please insert references next to study. Also, try perhaps collapsing some cells to avoid repetition (for mechanism, etc.) or perhaps divide table into segments by mechanism (since you seem to order studies by this) or similar to have it more compact, save space and make it easier to read. Also, it would be good to have the direction of the effect (positive, negative, etc.) in the findings column. It might be useful to state the city/province/state if available for a country where the study was conducted, since healthcare system might be organized differently if country is decentralized, like in case of Italy or USA, Swiss, etc.

Results - try rephrasing sentences where you number the studies, i.e. line 224 and later: “7 of 7 studies… 1 of 2 studies…” across the text, since this seems copy-pasted, monotonous for reading, try sum it up a bit. Perhaps try all studies (if 2 of 2), 50% of studies, majority of studies, etc. to make it more diverse and dynamic for readers. I highly appreciate that you divided results in segments by mechanism, much clearer now and easier to navigate.

Discussion – in line 279 was stated “Unlike previous reviews [13, 15, 19, 22], we found that PPR positively influenced consumers’ (i.e. patients, providers, purchasers) selection of healthcare providers (i.e. individual physician, hospital)..” while previously in results line 183 you wrote that 50% demonstrated positive effect: “In summary, 5 of 10 studies reported positive effects of PPR, with 1 of 10 a negative effect and 4 of 10 no effect. These findings suggest moderate level of evidence for PPR and selection of patients, physicians and hospitals.” This should be modified in the light of your findings. Also, some of the previous reviews did find positive effect (with smaller number of included studies in respect to here) on patients’ choice of surgeons, of healthcare plan, of nursing home, etc.

Discussion needs references insertion across text, i.e. lines 301-316 are completely lacking in references even though several hypothesis and examples are mentioned, these should be appropriately inserted.

Implications segment – I’m very pleased to read this part, it is very useful and significantly increased the quality and applicability of the research.

Funding - please state if the funder (Medibank Better Health Foundation) had some role in any of the steps during this research (design of the study, selection & data collection, interpretation, etc.).

7. PLOS authors have the option to publish the peer review history of their article (what does this mean?). If published, this will include your full peer review and any attached files.

Reviewer #1: No

---

## [Author Response · Author response to Decision Letter 1]

3 Dec 2020

Please see responses to reviewers document.

---

## [Decision Letter · Decision Letter 2]

22 Dec 2020

PONE-D-20-05627R2

Mechanisms and impact of public reporting on physicians and hospitals' performance: A systematic review (2000-2020)

PLOS ONE

Dear Dr. Prang,

Thank you for submitting your manuscript to PLOS ONE. After careful consideration, we feel that it has merit but does not fully meet PLOS ONE’s publication criteria as it currently stands. Therefore, we invite you to submit a revised version of the manuscript that addresses the points raised during the review process.

The manuscript greatly improved. Just some minor issues.

We look forward to receiving your revised manuscript.

Kind regards,

Lamberto Manzoli, M.D., M.P.H.

Academic Editor

PLOS ONE

Reviewers' comments:

Reviewer's Responses to Questions

**Comments to the Author**

1. If the authors have adequately addressed your comments raised in a previous round of review and you feel that this manuscript is now acceptable for publication, you may indicate that here to bypass the “Comments to the Author” section, enter your conflict of interest statement in the “Confidential to Editor” section, and submit your "Accept" recommendation.

Reviewer #1: (No Response)

2. Is the manuscript technically sound, and do the data support the conclusions?

Reviewer #1: Yes

3. Has the statistical analysis been performed appropriately and rigorously? 

Reviewer #1: N/A

4. Have the authors made all data underlying the findings in their manuscript fully available?

Reviewer #1: Yes

5. Is the manuscript presented in an intelligible fashion and written in standard English?

Reviewer #1: Yes

6. Review Comments to the Author

Reviewer #1: Thank you for considering previous suggestions and for modifying the manuscript.

Here are some minor changes to be conducted before the final decision is made:

• Not essential, but since last time it was also underlined by the Editor „quantitative estimates of effects should be reported. This will also help understanding whether it was really impossible to perform a meta-analysis, of it might have been performed on some specific outcomes.“ I feel it would be more appreciated to have quantitative estimates of the effect in a separate column (just numeric values) so the reader doesn’t have to search across the text in the column “Findings”, and so it would be much easier to understand and compare the effect among the studies, and also the possibility of conducting a meta-analysis.

• In lines 140-141 Authors say: „The methodological quality of each study was graded as very low, low, moderate or high (see S3 Appendix).“ But the numerical value for the cut off points to classify the Overall strength of evidence is not reported (i.e. how many stars were considered for the classification of high, moderate, etc.)?

• Let’s just say you did not considered studies with low quality in the synthesis (so we don’t create confusion) but still have extracted the data rather than excluding (ignoring) them, thus the sentence „Studies considered to be of low methodological quality were excluded from the synthesis (see S4 Appendix)“ should be modified somehow for reader to know that these information is still available in the Supplementary file (now when you wrote “see S4 Appendix” it seems like just the list of these studies is available) since usually when you say they are excluded you don’t extract data, so better to underline this. I also feel this sentence belong to the Results section now, since data was extracted (perhaps in lines 170-171?)

• Make sure that dissertations included (line 187) were peer-reviewed before making it available to the public since this might introduce bias?

Table 2 further suggestions:

• Please modify the title of the Table2 appropriately (i.e. characteristics, main findings or so...)

• Column „Country“ should be named also region/state/city or so...

• Type of PPR - try to define few main types of PPR to make it more uniform and easier to follow – in general, there are few main types of PPR (report cards, survey results, or so) and more details could be added in the parenthesis, i.e. Report Cards (Cleveland Health Quality Choice). Now it seems there are so many different types, which is not true. Also some types are not clear which type they are (i.e. Patients First (voluntary public reporting))?

• CABG acronym needs explanation below table

• Perhaps consider grouping columns „lever of reporting „ and „type of PPR“ into one column, since there are only few information on „level of reporting“ instead of repeating the information. Or some symbol (asterisk or so) can just be added for physician/hospital/clinic to save space and make table more appealing and clear.

• Level of reporting „village“ for Fabbri et al. is unclear? It refers to what? Since then in lines 169-170 you wrote: “Fabbri et al. [47] reported no effects of PPR on the selection of hospitals..”

• In column „Findings“ when you report „no effect“ - it means there was no statistically significant effect or there was no difference at all after the PPR? In case of non stat. sign. findings, the measures of effect should still be reported (even if not significant). This would still allow the possibility of conducting a meta-analysis in case of at least two studies with common outcome and effects reported.

• When reporting the effect, it should be uniformly presented in a consistent way (effect, 95% CI, p-value) it’s rather strange to not have them all reported in the original studies (i.e. OR should always be accompanied by 95% CI and p-value), please also check suppl. files of the original papers. This should be modified across the column „Findings“. Decimals should be always reported in the same way (rounded to two or three numbers after the comma)

• Also, as mentioned above, quantitative estimates of the effect would appeal better in a separate column.

Further comments:

• Line 225, these three studies (ref. 60, 61, 62) are “conducted on the same population” (this clarification should be added in some way next to the RCT to make it clear).

• Across the text, numbers less than 10 are usually written in words

• Since in the Acknowledgment you only mentioned Dr. Angela Zhang “for conducting risk of bias assessment and data extraction of studies from the third search” I was just wondering is the process of methodological quality evaluation conducted by just dr. Angela Zhang or was it conducted by two reviewers after which their evaluations were confronted, as you stated in lines 128-132 for screening titles and abstracts? Excluding studies based on the level of assessed methodological quality is a critical point of this review, and it must be conducted as objectively as possible in order not to introduce bias into your conclusions.

7. PLOS authors have the option to publish the peer review history of their article (what does this mean?). If published, this will include your full peer review and any attached files.

Reviewer #1: No

---

## [Author Response · Author response to Decision Letter 2]

28 Jan 2021

Please see response to reviewers file.

---

## [Decision Letter · Decision Letter 3]

5 Feb 2021

Mechanisms and impact of public reporting on physicians and hospitals' performance: A systematic review (2000-2020)

PONE-D-20-05627R3

Dear Dr. Prang,

We’re pleased to inform you that your manuscript has been judged scientifically suitable for publication and will be formally accepted for publication once it meets all outstanding technical requirements.

Kind regards,

Lamberto Manzoli, M.D., M.P.H.

Academic Editor

PLOS ONE

Additional Editor Comments (optional):

Reviewers' comments:

Reviewer's Responses to Questions

**Comments to the Author**

1. If the authors have adequately addressed your comments raised in a previous round of review and you feel that this manuscript is now acceptable for publication, you may indicate that here to bypass the “Comments to the Author” section, enter your conflict of interest statement in the “Confidential to Editor” section, and submit your "Accept" recommendation.

Reviewer #1: All comments have been addressed

2. Is the manuscript technically sound, and do the data support the conclusions?

Reviewer #1: Yes

3. Has the statistical analysis been performed appropriately and rigorously? 

Reviewer #1: N/A

4. Have the authors made all data underlying the findings in their manuscript fully available?

Reviewer #1: Yes

5. Is the manuscript presented in an intelligible fashion and written in standard English?

Reviewer #1: Yes

6. Review Comments to the Author

Reviewer #1: (No Response)

7. PLOS authors have the option to publish the peer review history of their article (what does this mean?). If published, this will include your full peer review and any attached files.

Reviewer #1: No

---

## [Editor Report · Acceptance letter]

10 Feb 2021

PONE-D-20-05627R3 

Mechanisms and impact of public reporting on physicians and hospitals’ performance:
A systematic review (2000-2020) 

Dear Dr. Prang:

I'm pleased to inform you that your manuscript has been deemed suitable for publication in PLOS ONE. Congratulations! Your manuscript is now with our production department. 

Kind regards, 

on behalf of

Dr. Lamberto Manzoli 

Academic Editor

PLOS ONE